# Prediction of the Weld Pool Stability by Material Flow Behavior of the Perforated Weld Pool

**DOI:** 10.3390/ma13020303

**Published:** 2020-01-09

**Authors:** Ruiqing Lang, Yongquan Han, Xueyu Bai, Haitao Hong

**Affiliations:** College of Materials Science and Engineering, No. 49 Ai Min Road, Xin Cheng Qu, Inner Mongolia University of Technology, Hohhot 010051, Inner Mongolia Autonomous Region, China; langruiqing2@163.com (R.L.);

**Keywords:** VPPAW, aluminum alloy 2219, arc pressure secondary compression, material flow behavior, weld pool stability

## Abstract

This article presents the application of a computational fluid dynamics (CFD) finite volume method (FVM) model for a thermo-mechanical coupling simulation of the weld pool used in variable polarity plasma arc welding (VPPAW). Based on the mechanism of the additional pressure produced through self-magnetic arc compression and the jet generated from mechanical plasma arc compression, and considering the influence of arc height and keyhole secondary compression on arc pressure, a three-dimensional transient model of variable polarity plasma arc (VPPA) arc pressure was established. The material flow behaviors of the perforated weld pools were studied. The results show that three kinds of flow behavior existed in the perforation weld pools and it is feasible to predict the weld pool stability by the material flow behaviors of the perforated weld pools. The weld pools can exist stably if the material flow in the bottom of the perforated weld pools can form confluences with moderate flow velocities of 0.45 m/s, 0.55 m/s and 0.60 m/s. The weld pools were cut when the material flowed downward and outward with the maximum velocity of 0.70 m/s, 0.80 m/s. When the maximum material flow velocity was 0.40 m/s, the weld pool collapsed downward under the action of larger gravity. The thermo-mechanical coupling model was verified by the comparison of the simulation and experimental results.

## 1. Introduction

Aluminum alloy 2219 (AA2219) has the characteristics of high specific strength, good mechanical properties at low and high temperatures, high fracture toughness and excellent stress corrosion resistance. It is one of the ideal materials for manufacturing large-scale space tanks [1]. Variable polarity plasma arc welding (VPPAW) has great application potential in the welding of aluminum alloys with thicknesses of 4–20 mm [2], which can be used to realize the double-sided formation of an aluminum alloy plate through single-sided welding without a forming groove, also known as the “zero defect” welding method [3,4,5]. However, the application of VPPAW in the manufacturing industry is restricted by its narrow process range and intense process sensitivity.

Han et al. [6,7] investigated the process characteristics and weld stability of aluminum alloy in VPPAW. However, most of the welding parameters were determined empirically, requiring numerous experiments for verification. With the development of computer science, numerical simulation can be used to understand the mechanisms and physical interactions of various welding processes [8]. Using stainless steel as the welding material, Wu et al. [9,10,11,12,13,14] developed a thermo-mechanical model to gain a better understanding of energy propagation and calculate the dynamic keyhole evolution and the fluid flow in the weld pool. Wu, Shinichi Tashino et al. [15,16] investigated the weld pool convection and keyhole formation mechanism in the keyhole plasma arc welding (PAW). The experimental methods of the above numerical studies were PAW. For VPPAW, Yang et al. studied the change of keyhole size and weld pool material flow with the change of heat input in the process of vertical welding on the premise of taking the average of the positive and negative polarity current [17,18]. Chen et al. [19,20,21] mainly studied the physical properties of 5A06 and A5052P aluminum alloy during VPPAW.

Therefore, numerical simulation analysis regarding to the material flow behavior of AA2219 weld pool has seldom been carried out. However, AA2219 has a high thermal conductivity, so it dissipates heat quickly in the welding process, and the keyhole of AA2219 is more hardly formed than that of steel or 5-Series aluminum alloy in VPPAW [15]. Therefore, it is important to study the material flow behavior of AA2219 during VPPAW.

VPPAW essentially belongs to perforated welding, and thus, the keyhole evolution is extremely important. Generally, the keyhole is formed in several seconds, which is proposed to be the dominant driving force of arc pressure [15]. The arc pressure model established at present mainly considers the change of current and the arc pressure distribution along the thickness direction of the workpiece [9,10,19,22]. For VPPAW, the plasma gas flow rate, nozzle diameter, and welding speed have important impacts on the welding process. Thus, an arc pressure model affected by multiple welding parameters [23] was established based on the continuous plasma impact theory. However, the additional pressure generated by arc self-magnetic compression was not considered. Guo [24] pointed out that when the additional pressure changes along the arc length direction, the medium in the arc column driven by the axial pressure gradient flowing from high to low pressure impacts the workpiece, which cannot be neglected.

In our study, an improved three-dimensional transient arc pressure model affected by multiple parameters was developed based on the principle of additional pressure produced through self-magnetic arc compression and the jet generated by plasma arc compression. In addition, the model considered the influence of arc height and keyhole secondary compression on the arc pressure, which firstly decreased linearly with the increase in the arc height and increased squarely with the increase in the keyhole depth after the keyhole formation. The influence of the perforated weld pool flow behaviors on the weld pool stability was investigated. The simulation results were verified by experiments. The weld pool stability, predicted by the material flow behavior of the perforated weld pool, is feasible and the application of the computational fluid dynamics (CFD) finite volume method (FVM) model for a material flow behavior simulation for VPPAW is practical in formulating and improving the welding technology during production.

## 2. Experimental Methods

The VPPA welding and data acquisition system was schematically shown in Figure 1, which consisted of a welding power source (VPPA-2), plasma arc welding torch (PAW-300), workpiece, high-speed camera (Red Lake Y4) and an industrial computer.

The workpiece employed in this study was an AA2219 plate with a chemical composition as shown in Table 1.

The dimensions of the workpiece was 120 mm × 60 mm × 6 mm (length × width × thickness). The shielding gas used was pure argon with a volume fraction of 99.99% throughout the experiments. The high-speed camera technology was used to capture a real-time evolution image of the back keyhole.

The welding parameters are summarized in Table 2 [6,21]. To reduce the burning loss of a tungsten electrode, a large $I_{\mathrm{DCEP}}$ and a short $t_{\mathrm{DCEP}}$ are suggested to satisfy high energy while reducing the tungsten electrode loss and cleaning the oxidation film [26].

## 3. Model Description

The model was developed in an Ansys–Fluent environment based on FVM, which allowed the simulation of the VPPA welding process to be conducted.

### 3.1. Basic Assumptions

(1) Molten metal is considered as a laminar incompressible Newtonian fluid affected by the plasma arc pressure, surface tension, electromagnetic force and gravity [9].

(2) A Boussinesq approximation was used to treat the buoyancy term [27].

(3) The thermal conductivity, specific heat, surface tension and viscosity are related to temperature, whereas the other thermo-physical parameters are constant [18].

### 3.2. Governing Equations

The continuous equation, momentum conservation equation, and energy conservation equation need to be solved to obtain the perforation process and the weld pool flow behavior for VPPAW.

#### 3.2.1. Continuous Equation

The continuous equation can be expressed through Equation (1):(1)$$\partial \rho /\partial t+\nabla \cdot \rho \vec{v}=0$$
where ρ is the constant fluid density, *t* is the time, $\vec{v}$ is the velocity vector.

#### 3.2.2. Momentum Conservation Equation

The momentum conservation equation in the *x*, *y*, *z* directions can be written as follows:(2)$$\rho \left[\frac{\partial u}{\partial t}+u\frac{\partial u}{\partial x}+v\frac{\partial u}{\partial y}+w\frac{\partial u}{\partial z}\right]=-\frac{\partial p}{\partial x}+\mu \left(\frac{\partial^{2}u}{\partial x^{2}}+\frac{\partial^{2}u}{\partial y^{2}}+\frac{\partial^{2}u}{\partial z^{2}}\right)+S_{x}$$
(3)$$\rho \left[\frac{\partial v}{\partial t}+u\frac{\partial v}{\partial x}+v\frac{\partial v}{\partial y}+w\frac{\partial v}{\partial z}\right]=-\frac{\partial p}{\partial x}+\mu \left(\frac{\partial^{2}v}{\partial x^{2}}+\frac{\partial^{2}v}{\partial y^{2}}+\frac{\partial^{2}v}{\partial z^{2}}\right)+S_{y}$$
(4)$$\rho \left[\frac{\partial w}{\partial t}+u\frac{\partial w}{\partial x}+v\frac{\partial w}{\partial y}+w\frac{\partial w}{\partial z}\right]=-\frac{\partial p}{\partial x}+\mu \left(\frac{\partial^{2}w}{\partial x^{2}}+\frac{\partial^{2}w}{\partial y^{2}}+\frac{\partial^{2}w}{\partial z^{2}}\right)+S_{z}$$
where *u*, *v*, and *w* are the velocities in *x*, *y*, and *z* directions, respectively; ρ is the mixture density; and μ is the viscosity. The momentum source terms $S_{x}$, $S_{y}$, and $S_{z}$ in Equations (2)–(4) can be expressed as follows:(5)$$S_{x}=A_{x}+F_{x}$$
(6)$$S_{y}=A_{y}+F_{y}$$
(7)$$S_{z}=A_{z}+F_{z}+P_{a}\left(x,y\right)$$
where $A_{x}$, $A_{y}$, and $A_{z}$ are the momentum loss in the solid–liquid two-phase region caused by solidification in the *x*, *y* and *z* direction, respectively; $F_{x}$, $F_{y}$, and $F_{z}$ are the electromagnetic force in the x, y and z direction, respectively; and $P_{a}$ (x,y) is the arc pressure, which is described individually in Section 3.4.

The momentum loss in the solid–liquid two-phase region is described as follows:

In the solidification process of the weld pool, the amount of flowing liquid metal gradually decreases to zero, which leads to the momentum loss of the flowing metal. An enthalpy–porosity technique is used to deal with the momentum loss of a solid–liquid zone [28] (mushy zone). The enthalpy–porosity technique is essentially used to calculate the momentum loss in each element based on temperature. To be specific, the liquefaction volume fraction of the unit is calculated based on the temperature of the unit. The momentum loss from solidification is calculated through Equation (8):(8)$$A=A_{mush}\frac{{\left(1-\beta \right)}^{2}}{\left(\beta^{3}+\xi \right)}\vec{v}$$

Here, $A_{x}$, $A_{y}$, and $A_{z}$ obtained through the orthogonal decomposition of Equation (8) on the *x*, *y*, and *z* axes are as follows:(9)$$A_{x}=\left[-\frac{A_{mush}{\left(1-\beta \right)}^{2}u}{\left(\beta^{3}+\xi \right)}\right]$$
(10)$$A_{y}=\left[-\frac{A_{mush}{\left(1-\beta \right)}^{2}v}{\left(\beta^{3}+\xi \right)}\right]$$
(11)$$A_{z}=\left[-\frac{A_{mush}{\left(1-\beta \right)}^{2}w}{\left(\beta^{3}+\xi \right)}\right]$$
where $A_{mush}$ is the mushy zone constant, which is used to adjust the damping coefficient; $\vec{v}$ is the velocity vector of the fluid, and β is the volume fraction of the liquid phase and can be expressed through the following equation.
(12)$$\beta =\{\begin{array}{c} 0\;,T\leq T_{s}\\ 1,\;T\geq T_{l}\\ \left(T-T_{s}\right)/(T_{l}-T_{s}),T_{s}<T<T_{l} \end{array}$$

According to Maxwell’s electromagnetic theory, the electromagnetic forces [29] are expressed in a Cartesian coordinate system as follows:(13)$$F_{x}=-\frac{\mu_{0}I^{2}}{4\pi^{2}\sigma_{j}^{2}r}\;exp\left(-\frac{r^{2}}{2\sigma_{j}^{2}}\right)\left[1-exp\left(-\frac{r^{2}}{2\sigma_{j}^{2}}\right)\right]{\left(1-\frac{z}{L}\right)}^{2}\frac{x}{r}$$
(14)$$F_{y}=-\frac{\mu_{0}I^{2}}{4\pi^{2}\sigma_{j}^{2}r}exp\left(-\frac{r^{2}}{2\sigma_{j}^{2}}\right)\left[1-exp\left(-\frac{r^{2}}{2\sigma_{j}^{2}}\right)\right]{\left(1-\frac{z}{L}\right)}^{2}\frac{y}{r}$$
(15)$$F_{z}=\frac{\mu_{0}I^{2}}{4\pi^{2}Lr^{2}}{\left[1-exp\left(-\frac{r^{2}}{2\sigma_{j}^{2}}\right)\right]}^{2}\left(1-\frac{z}{L}\right)$$
where $r=\sqrt{{\left(x-vt\right)}^{2}+y^{2}}$ and I is the welding current, which is described in Equation (17).

#### 3.2.3. Energy Conservation Equation

According to the energy conservation code, the increased rate of energy in the microelement is equal to the net heat flux in the microelement, added to the mass force and the work conducted on the surface of the microelement, which can be expressed as follows [30]:(16)$$\rho \left[\frac{\partial H}{\partial t}+u\frac{\partial H}{\partial x}+v\frac{\partial H}{\partial y}+w\frac{\partial H}{\partial z}\right]=\frac{\partial }{\partial x}\left(k\frac{\partial T}{\partial x}\right)+\frac{\partial }{\partial y}\left(k\frac{\partial T}{\partial y}\right)+\frac{\partial }{\partial z}\left(k\frac{\partial T}{\partial z}\right)+S_{H}+S_{V}$$
where $S_{H}=h+\Delta H$; $h=h_{ref}+\int_{T_{ref}}^{T}C_{P}dT$; $\Delta H=\beta \;L_{m}$. $S_{H}$ is the energy change caused by the liquid phase transition; $h_{ref}$ is the reference enthalpy; $T_{ref}$ is the reference temperature; $C_{P}$ is the specific heat; β is the volume fraction of the liquid phase, which can be expressed through the Equation (12), and $L_{m}$ is the latent heat of fusion. $S_{v}$ is the heat source equation and can be expressed as Equations (18) and (19).

### 3.3. Energy Input for VPPAW Process

The welding current in VPPAW, which changes in stages, has a significant impact on energy input into the weld, and thus the current expression should first be determined. As shown in Figure 2 below, the current change stages include the preheating stage, current rising stage, variable polarity stage, and current descending stage [31].

The current variation in Figure 2 is represented through Equation (17):(17)$$\{\begin{array}{c} I=I_{0,}0\leq t\leq t_{0};\\ I=I_{0}+\left(t-t_{0}\right)*tan\alpha ,t_{0}\leq t\leq t_{S};\\ I=\sqrt{(I_{DCEP}{}^{2}*t_{DCEP}+I_{DCEN}{}^{2}*t_{DCEN})/T_{c}};\\ t_{S}\leq t\leq t_{M};\\ I=I_{0}-\left(t-t_{0}\right)*tan\beta ,t_{M}\leq t\leq t_{N}; \end{array}$$
where *tan α* = ($I_{DCEN}-I_{0}$)/(*t_S_*
−
*t*_0_); tan *β =*
$I_{DCEN}-\left(t_{N}-t_{M}\right)$; $I_{0}$ is the preheating current, *t*_0_ is the preheating time; $t_{S}$ is the current rising time from the beginning of the welding process to $I_{DCEN}$; $t_{M}$ is the time from the beginning of the welding process to the start of the keyhole closure phase; $t_{N}$ is the total welding time; and $T_{C}$ is the positive and negative polarity current cycle time during the variable polarity stage.

The heat transfer model $S_{v}$ was constructed and consists of a double ellipsoid heat source on the top and a conical heat source below [32].

The heat flux distribution of the double ellipsoid heat source at the (*x*, *y*, *z*) points is expressed as follows:(18)$$\{\begin{array}{c} q_{f}=\frac{12\sqrt{3}U{Im}_{1}f_{1}}{\pi \sqrt{\pi }a_{f}bc}{exp}^{\left(\frac{-3^{2}}{a_{f}{}^{2}}-\frac{3y^{2}}{b^{2}}-\frac{3z^{2}}{c^{2}}\right)}\;(x\geq 0,z\geq -0.003)\\ q_{r}=\frac{12\sqrt{3}U{Im}_{1}f_{2}}{\pi \sqrt{\pi }a_{r}bc}{exp}^{\left(\frac{-3^{2}}{a_{r}{}^{2}}-\frac{3y^{2}}{b^{2}}-\frac{3z^{2}}{c^{2}}\right)}\;(x\geq 0,z\geq -0.003) \end{array}$$
where *m*_1_ is the distribution coefficient of the double ellipsoid heat source, $f_{1}$ and $f_{2}$ are the distribution coefficients of the front and rear heat inputs of the double ellipsoid heat source, and $f_{1}+f_{2}=1$.

The heat flux distribution of the conical heat source at the (x, y, z) points is expressed as follows [33]:(19)$$q_{v}=q\left(x,y,z\right)=\frac{9m_{2}IUe^{3}}{\pi H\left(e^{3}-1\right)\left(r_{e}^{2}+r_{e}r_{i}+r_{i}^{2}\right)}exp\left(-\frac{3x^{2}+3y^{2}}{r_{0}^{2}\left(z\right)}\right)\;(-0.006\leq z<-0.003)$$
where $r_{0}\left(z\right)=r_{e}-\left(r_{e}-r_{i}\right)\frac{z_{e}-z}{z_{e}-z_{i}}$; $r_{e}$ and $r_{i}$ are the radius of the conical heat source for the top and bottom workpiece surfaces; $z_{e}$ and $z_{i}$ are the *z* component coordinates for the top and bottom workpiece surfaces, respectively; m_2_ is the distribution coefficient of the conical heat source, and *m*_1_
+
*m*_2_ = 1.

### 3.4. Arc Pressure for VPPAW Process

The arc pressure is an important factor in the keyhole evolution. Based on the mechanism of additional pressure produced through arc self-magnetic compression and a jet generated by plasma arc mechanical compression, a three-dimensional transient arc pressure model was developed. In addition, considering that the arc pressure first decreases with the increase in the arc height, and then increases with the additional constraints of the keyhole after the keyhole formation, the arc pressure model is further optimized. The plasma arc pressure can be expressed as follows [34]:(20)$$P=P_{0}+P_{v}$$
where *P* is the total arc pressure, $P_{0}$ is the static arc pressure, and $P_{v}$ is the dynamic arc pressure.

#### 3.4.1. Static Arc Pressure

The arc column expanding from the nozzle to the workpiece is assumed to have a conical shape, and the current distribution along the solid angle of the cone is uniform. A schematic diagram of the electromagnetic static arc pressure distribution is shown in Figure 3.

The static arc pressure at any position in the arc column (ι, ψ) can be expressed through the following Equation (21) [35]
(21)$$P_{0}^{\prime }=\frac{2I^{2}}{\pi l^{2}{\left(1-cos\theta \right)}^{2}}\log^{\frac{\cos\left(\psi /2\right)}{\cos\left(\theta /2\right)}}\;(\mathrm{dyn}/{\mathrm{cm}}^{2})$$
where *I* is the welding current calculated through Equation (17), θ is half of the conical angle, ψ is the angle between the *z*-axes and the axis of observation point A, and l is the distance (cm) between point A and the vertex of the conical angle.
$$P_{0}^{\prime \prime }=K^{\prime }\times \frac{0.2I^{2}}{\pi l^{2}{\left(1-cos\theta \right)}^{2}}\log^{\frac{1}{\cos\left(\theta /2\right)}}\;(N/m^{2})$$

The static arc pressure at the center of the plasma arc is the largest when l is a constant. Namely, the maximum static arc pressure $P_{0}^{\prime \prime }$ can be obtained if ψ=0 and l is the distance from the torch to the workpiece. *K’* is an adjusting coefficient which is introduced to consider the influence of the change in plasma gas flow rate on θ. Assuming that the workpiece under the arc is completely in a melting state, the molten pool will be concave [24] under the static arc pressure.

#### 3.4.2. Dynamic Arc Pressure

The distribution of energy and force at the interface between the keyhole and the weld pool is fatal for the keyhole formation and maintenance [9,20]. During the VPPAW process, the effects of the plasma jet promote the keyhole penetration. The density of the plasma arc is approximately the same. According to Bernoulli’s theorem, the dynamic arc pressure $P_{v}$ is obtained as follows:(22)$$P_{v}=\rho V^{2}/2$$
where *V* is the velocity of the plasma jet, ρ is the plasma density, and $P_{v}$ is the pressure difference between Sections A and B.

Figure 4 shows the internal structure of the nozzle. We assume that the plasma in the nozzle satisfies the law of mass conservation of a fluid flow, $\rho_{A}V_{A}A=\rho_{B}V_{B}B$, where $V_{A}$, $\rho_{A}$, and A, and $V_{B}$, $\rho_{B}$, and B, are the velocity of the plasma flow, density, and area in sections A and B, respectively.

After igniting the plasma arc, the velocity in Section B is calculated through the following equation:(23)$$V_{B}=(Q/B)\left(T_{B}/T_{A}\right)\left(M_{A}/M_{B}\right)\left(P_{A}/P_{B}\right)\left(1+\alpha \right)$$
where $M_{A}$ and $M_{B}$ are the plasma molecular weight flowing through Sections A and B, respectively; B is the section area at the nozzle outlet; $P_{A}$, $T_{A}$, and $\rho_{A}$, and $P_{B}$, $T_{B}$, and $\rho_{B}$, are the dynamic pressure, plasma temperature, and plasma density in Sections A and B, respectively; Setting $M_{A}$ = $M_{B}$ = 39.95 g/mol and $P_{A}$ = $P_{B}$ = 1.01 × 10^5^ N/m^2^. $T_{A}$ = 3000 K, $T_{B}$ = 18,000 K [36], and $K^{\prime \prime }$ is an adjusting coefficient which is introduced to consider the influence of the change in plasma gas flow rate on the ratio of $T_{B}/T_{A}$. $V_{B}$ can be obtained through Equation (24):(24)$$V_{B}=\left[K^{\prime \prime }\times \left(T_{B}/T_{A}\right)\right](Q/B)\left(1+\alpha \right)$$
where α is a adjusting coefficient, which was treated as a constant. Introducing $V_{B}$ into Equation (24), the dynamic arc pressure at the nozzle outlet can be obtained as follows:(25)$$P_{v}^{\prime }=\frac{1}{2}\rho_{Ar}{\left(\frac{Q}{\pi {\left(d/2\right)}^{2}}\times \frac{K^{\prime \prime }\times T_{B}}{T_{A}}\times \left(1+\alpha \right)\right)}^{2}$$

Because the velocities in the arc axes decrease gradually along the downstream direction, owing to the additional pressure, the gradient decreases from the nozzle to the weld pool surface. Here, $V_{B}$ can be considered as the maximum velocity in the arc column. We assume that the plasma density $\rho_{Ar}$ is a constant. The velocity of the nozzle outlet is approximately equal to that of the workpiece surface, and thus $P_{v}^{\prime }$ can be regarded as the largest dynamic pressure on this surface.

To summarize, in VPPAW, the total arc pressure *P* can be obtained as follows: the arc pressure inside the workpiece conforms to a Gaussian distribution [23]:(26)$$P=(\frac{K^{\prime }2I^{2}}{\pi L^{2}{\left(1-\cos\theta \right)}^{2}}\log^{\frac{1}{\cos\left(\theta /2\right)}}+\frac{1}{2}\rho_{Ar}\left(\frac{Q}{\pi {\left(d/2\right)}^{2}}\times \frac{K^{\prime \prime }\times T_{B}}{T_{A}}\times \left(1+\alpha \right){)}^{2}\right)\times exp^{\left(\frac{-3r^{2}}{r_{p}{}^{2}}\right)}$$
where *d* is the nozzle diameter; *r* is the radius of an arbitrary cross section; $r_{p}$ is the radius of the plasma arc force impacting on the workpiece. Significantly, in the above arc pressure model, the current varies according to the different loading stages.

#### 3.4.3. Arc Pressure Model Improvement

The plasma arc pressure was decreased at first due to the increase in the arc length, and then increased due to the additional constraint of the keyhole [20]. For simplicity, in this study, from the top to the one-sixth of the base metal, the arc pressure is assumed to be linearly decreased as Equation (27), and the arc pressure increases quadratically along the direction of the workpiece thickness, as Equation (28) from the keyhole appearance to penetration. Based on Pan′s study [21], the keyhole begins to appear about 1 mm below the workpiece. The improved arc pressure model P′ can expressed as following:$$P^{\prime }=P\times C_{\mathrm{arc}}$$
(27)$$C_{\mathrm{arc}}=1+\varphi \times Z_{k}\;(-1\mathrm{mm}\leq Z_{k}<0\;\mathrm{mm})$$
(28)$$C_{\mathrm{arc}}=1+\varphi {\left(\frac{h_{\mathrm{kh}}}{L}\right)}^{2}\;\;\;\;\left(-6\;\mathrm{mm}\leq h_{\mathrm{kh}}<-1\;\mathrm{mm}\right)$$
where φ is more than zero and less than one and is introduced to consider the variation of the arc pressure with the change of weld pool depth. $Z_{k}$ is the weld pool depth, $h_{kh}$ is the real-time keyhole depth and L is the workpiece thickness. It was found that φ equal to 0.4 was suitable throughout the tests.

## 4. Simulation Consideration

### 4.1. Computational Domain

The computational domain for the VPPAW model is shown in Figure 5. The domain is symmetric about the *x*–*z* plane. To improve the computational efficiency, the length (*x*-axes), width (*y*-axes) and height (*z*-axes) of the workpiece are 30 mm, 15 mm, and 10 mm, respectively. The domain is divided into three layers, the middle of which is a 6-mm fluid layer. The upper and lower layers are 2-mm air layers. The minimum grid size is 0.3 mm.

### 4.2. Keyhole Tracking

(29)$$\partial F/\partial t+(\nabla \cdot \vec{v)}F=0$$

A two-phase (gas and aluminum alloy) flow model was used [37]. In this case, volume fraction function *F* = 1 corresponds to cells full of metal fluid, while *F* = 0 corresponds to cells empty of metal fluid. Cells with F lying between 0 and 1 locate in the free surface.

### 4.3. Boundary Conditions

Figure 6 shows a schematic diagram of the boundary conditions, which were set as indicated in Table 3 below.

The surface tension of the gas-liquid interface is adopted using the continuum surface force (CSF) model [38] and calculated according to the following equation [39].
(30)$$\gamma =\{\begin{array}{c} 0.825\;\left(T=T_{s}\right)\\ 0.825-0.5\times 10^{-4}T\;(T_{l}<T<T_{b}) \end{array}$$
where *T* is the real-time temperature of the weld pool.

### 4.4. Properties of AA2219

Some thermo-physical properties of AA2219 were set out in Table 4, as follows:

For AA2219, the temperature dependence of density is represented by the following linear equation [39]:(31)ρ(T)=(2904)+(−0.2603+0.0069)×T

The specific heat capacity and viscosity of AA2219 are set through the following equations [17]:(32)$$C_{p}\left(T\right)=338.98+1.359T-8.027\times 10^{-4}T^{2}$$
(33)$$\mu =4.62\times 10^{-3}-5.49\times 10^{-6}T+1.92\times 10^{-9}T^{2}$$

The thermal conductivity of AA2219 is calculated by the following Equation [38]:(34)$$k\left(T\right)=\{\begin{array}{c} 123.38+0.048T-1.018\times 10^{-4}T^{2}\;\left(T\leq T_{l}\right)\\ \frac{3R_{0}}{\sqrt[{3}]{N_{0}}}{\left[\frac{R_{0}T_{0}}{M}\left(101.6-82.4\frac{T}{T_{b}}\right)\right]}^{1/2}{\left(\frac{\rho }{M}\right)}^{2/3}(T_{l}<T<T_{b}) \end{array}$$

In Equations (31)–(33), *T* is the temperature of the weld pool. In Equation (34), $R_{0}$ is the ideal gas constant, $N_{0}$ is the Avogadro constant, and M is the molar mass of aluminum.

## 5. Results and Discussion

Firstly, the numerical results of case A were analyzed, including the dynamic change of the keyhole size, the arc pressure distribution on the keyhole surface, the temperature field and the flow behavior of the molten metal in the weld pool.

Figure 7 shows the arc pressure distribution during the keyhole evolution for case a. In Figure 7a–h, W is half of the keyhole width on the upper workpiece surface, and D is the keyhole depth. The maximum arc pressure appears at the bottom of the keyhole, reaching 3000 Pa. Jiang et al. [40] found that the arc pressure is between 2500 and 4500 Pa, and thus the arc pressure calculated in the present study is reasonable. At 1.72 s, a small pool is formed with a half width of 1.2 mm and a half depth of 0.7 mm. Based on the thermo-mechanical coupling effect, at 2.43 s, the arc pressure penetrates into the weld pool to form a keyhole with a half width of 2.0 mm on the upper surface and a depth of 1.8 mm. Subsequently, the keyhole size increases continuously, and the increasing rate of the keyhole depth is greater than that of the keyhole width. This is because of the secondary compression of the keyhole to the arc pressure. The workpiece is penetrated at 4.55 s.

Figure 8 shows the material flow and temperature distribution during the keyhole evolution for case a. The arrows represent the directions of the material flow. At first, the workpiece surface is heated by the plasma and depressed by the arc pressure. As the heat accumulates, at 1.72 s, the molten metal forms eddies, as shown in Figure 8a ① ②, under the combined action of the arc pressure, surface tension, and electromagnetic force. The eddies last until 3.65 s. The velocity at the interface between the keyhole and the weld pool is the largest. At 4.34 s, the molten metal in the lower part of the weld pool flows upward and outward, as shown in Figure 8e ①, which accelerates the keyhole penetration, due to the obvious compression effect of the keyhole on the arc pressure and the digging role of the arc pressure is enhanced. At 4.55 s, the molten metal in the lower weld pool converges towards the center, as shown in Figure 8f ①.

The simulation results show that the maximum arc pressure during the DCEN and DCEP phase are 2600 Pa and 3000 Pa as shown in Figure 9a,b respectively.

In order to verify the accuracy of the arc pressure model, an arc pressure measure and data acquisition system were set up schematically, as shown in Figure 10.

The system consisted of a welding power source (VPPA-2), a plasma arc welding torch (PAW-300), a pressure transmitter (YB005-01) developed by General Engineering Research Institute of China Academy of Sciences, an Agilent memory oscilloscope (6422D) and an industrial computer data acquisition card (PCL1800) produced by Advantech. The welding power source was the self-developed VPPA-2 variable polarity plasma inverter with an 80C196KC single chip microcomputer as the control core and the main circuit as the double inverse circuit topological structure. The plasma arc pressure was transmitted to the pressure transmitter through the copper plate with the keyhole and the copper tube. The output voltage signal of the pressure transmitter was collected synchronously by the industrial computer and the memory oscilloscope.

The signal converter converts the voltage signals into digital signals. The waveform measured by the oscilloscope corresponds to the arc pressure. As Figure 11 and Figure 12 show, the vertical voltage axis is 2000 Pa/grid, and the horizontal time axis is 30 ms/grid. The waveform shows that the average values of the arc pressure during the DCEN phase and DCEP phase are about 2400 Pa and 2700 Pa, respectively. The measured results are consistent with the simulation results, and the accuracy of the arc pressure model is verified.

The calculated keyhole diameter is 1.86 mm, which is consistent with the measured value of 1.82 mm.

The calculated fusion line (FL) is consistent with the measured fusion line obtained from the experiment, as shown in Figure 13.

For case b–f, simulation tests and welding experiments were successively carried out with the welding parameters in Table 2. The comparison results in Table 5 show that the calculated results were in good agreement with the experimental results and the accurate thermo-mechanical coupling model lays the foundation for the subsequent results analysis.

For case a–f, the simulation results of the material flow of the perforated weld pool and phase distribution are shown in Figure 14, and there are three kinds of flow behavior. Since the temperature gradients inside the weld pools, as shown in Figure 15, have significant impacts on the material flow behaviors of the perforated weld pools, the two figures will be discussed relevantly below.

For cases a–f, the temperature gradient inside the weld pool was studied to further explore the mechanism of the weld pool material flow behavior to weld pool stability. The temperature value of the z = −5 mm straight line on the xoz plane was extracted. Since the weld pool is symmetrical, the temperature value within 3.5 mm (from the keyhole center to its right side) is shown in Figure 15.

Due to the different horizontal distances from the weld pool boundary to the *x*-axis for cases a–f, the boundaries of the perforated weld pool were shifted to x = 6 mm in order to compare the slope of the temperature gradients. As shown in Figure 15b, the slope of case d was the largest, the angle between the slope line of cases a–b and the x negative direction horizontal line are 35° (Tan (35°) = 0.70) and 41° (Tan (41°) = 0.87), respectively, and the slope of case f was the smallest.

In Figure 14a–f, under the action of the surface tension and the digging effect of the arc pressure, the upper molten metal of the weld pool flows outwards and then forms clockwise, as shown in Figure 14a ①; The molten metal in the middle of the weld pool flows upwards and inwards, as shown in Figure 14a ②, under the digging action of the arc pressure, the electromagnetic force, and the effect of clockwise eddy flow in Figure 14a ①; in particular, the material flow behavior in the lower part of the weld pool is quite different. In case a, case c and case e, the molten metal forms confluences towards the keyhole center, as shown in Figure 14a ③, at the maximum velocities of 0.45 m/s, 0.55 m/s and 0.6 m/s, respectively, with moderate temperature gradients (the slopes values being greater than 0.7 and less than 0.87), as shown in Figure 15b. In case b and case d, the molten metal of the weld pool bottom flows downwards and outwards, as shown in Figure 14b ①, at the maximum speed of 0.7 m/s and 0.8 m/s, respectively, under the action of the large surface tension caused by the relatively large temperature gradients (slope value ≥ 0.87) in the weld pool, as shown Figure 15b. In case f, the molten metal flows as shown in Figure 14f ① ②, which makes the molten metal of the weld pool increase under the joint action of small arc pressure and surface tension. Although the weld pool metal can still form the confluence, as shown in Figure 14 f ③, the weld pool with a large amount of molten metal collapsed under the action of gravity.

In order to summarize the simulation results, according to the material flow behavior in the lower part of the weld pool, the equilibrium coefficient ƞ is introduced.
$$ƞ=\frac{Q}{I}$$
where Q is the plasma gas flow rate, its unit being mL/s.

When 1/4 ≤ ƞ < 1/3, such as in case a, case c and case e, the material flows to form confluences towards the keyhole center with the moderate velocities (0.45 m/s, 0.55 m/s and 0.6 m/s), and the weld pool can exist stably. When ƞ ≥ 1/3, such as in case b and case d, the material flows downwards and outwards with the maximum velocities of about 0.7 m/s or 0.8 m/s, and the weld pools are cut. When ƞ < 1/4, the maximum downward flow velocity of the perforated weld pool was 0.4 m/s and the weld pool with too large an amount of molten metal, caused by the long-time metal refluxing phenomenon, collapsed downward under the action of a larger gravity.

Figure 16a–c are the weld formations obtained from the experiments classified by ƞ. When ƞ ≥ 1/3 or ƞ < 1/4, the welds were cut or collapsed, as shown in Figure 16a,c, respectively, caused by the weld pool instability. When 1/4 ≤ ƞ < 1/3, the weld pools can stably exist and the welds can form well, as shown in Figure 16b.

The different weld formations further verify that it is feasible to predict the weld pool stability by the material flow behavior of the weld pool.

## 6. Conclusions

(1) It is feasible to predict the weld pool stability by the material flow behavior of the weld pool. The weld pool can exist stably when the material in the bottom of the weld pool can form confluences with the maximum flow velocities of 0.45 m/s, 0.55 m/s and 0.60 m/s. The weld pools are cut when the material flows downward and outward with the maximum velocities of 0.7 m/s and 0.8 m/s, and metal gaps appear according to the law of mass conservation. When the maximum material flow velocity is 0.40 m/s, the metal refluxing phenomenon occurs in the weld pool for a long time, increasing the amount of the molten metal. The gravity of the weld pool increases with the amount of molten metal, so the weld pool collapses downward under the action of a larger gravity.

(2) Based on the mechanism of the additional pressure produced through self-magnetic arc compression and the jet generated from mechanical plasma arc compression, and meanwhile considering that the arc pressure first decreases with the increase in arc height, and then increases with the additional constraints of the keyhole after the keyhole formation, a three-dimensional transient model of the VPPA arc pressure which first decreases linearly with the increase in arc height and then increases squarely with the increase in keyhole depth was established.

(3) The measured arc pressure value, the shape of the fusion line, and the keyhole size on the back of the weld were in good agreement with the simulation results.

## Figures and Tables

**Figure 1 materials-13-00303-f001:**
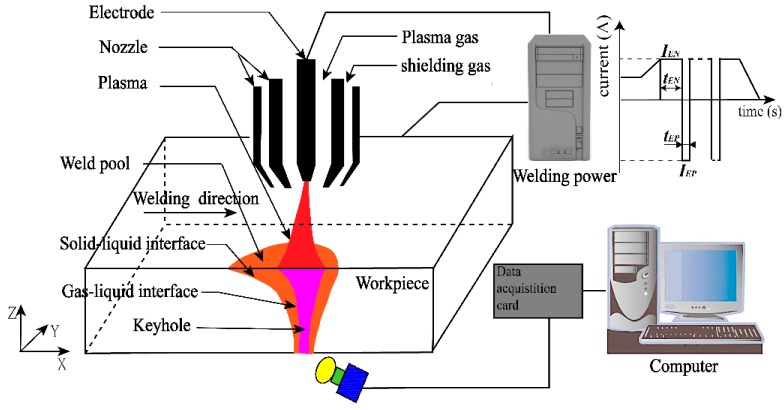
Schematic of variable polarity plasma arc (VPPA) welding and data acquisition system.

**Figure 2 materials-13-00303-f002:**
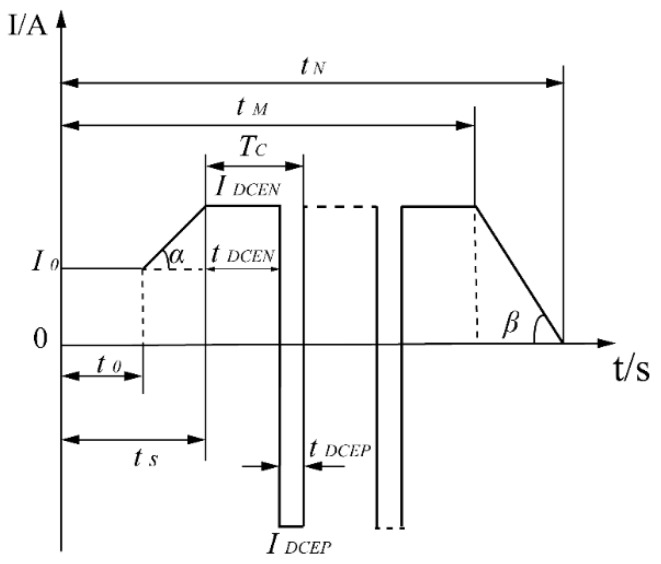
Schematic diagram of current variation during the VPPAW process.

**Figure 3 materials-13-00303-f003:**
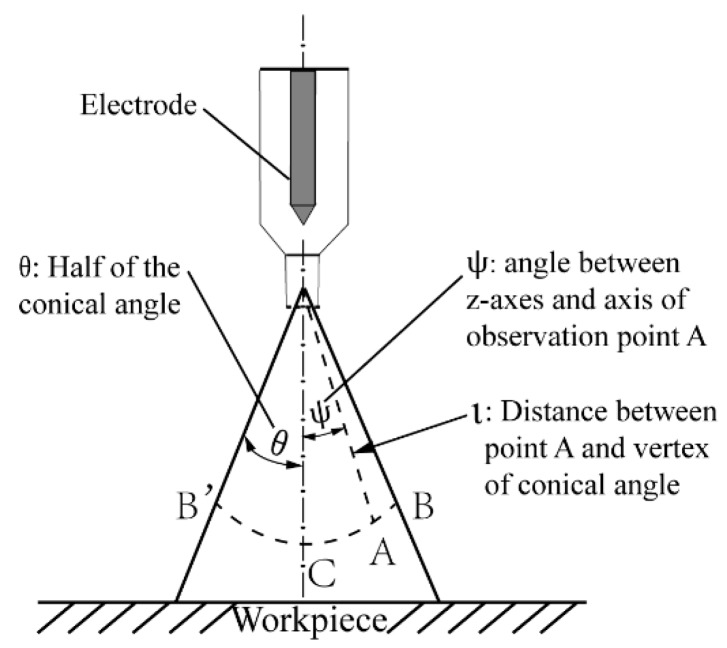
Schematic diagram of electromagnetic static arc pressure distribution.

**Figure 4 materials-13-00303-f004:**
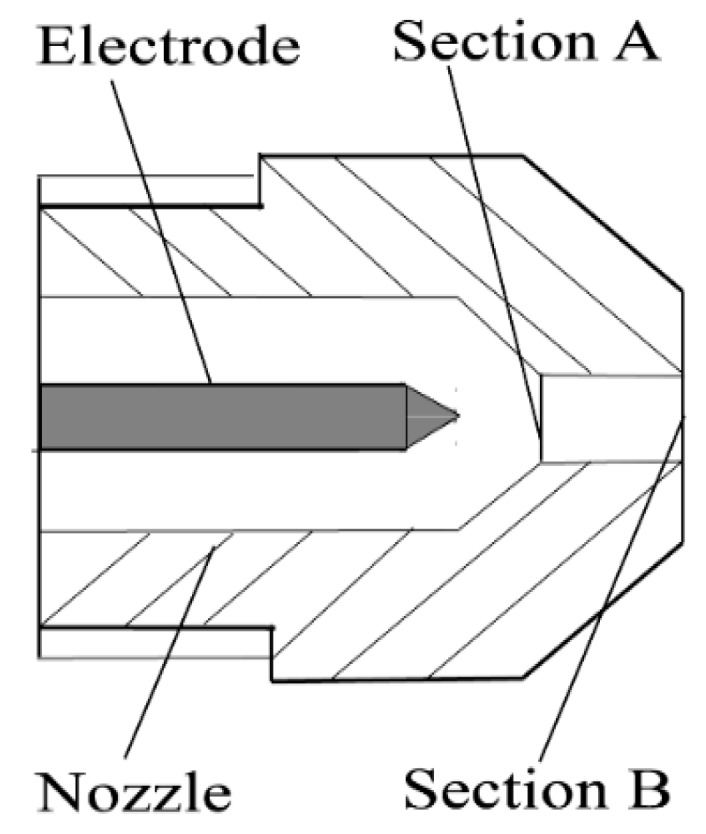
Internal structure of the nozzle.

**Figure 5 materials-13-00303-f005:**
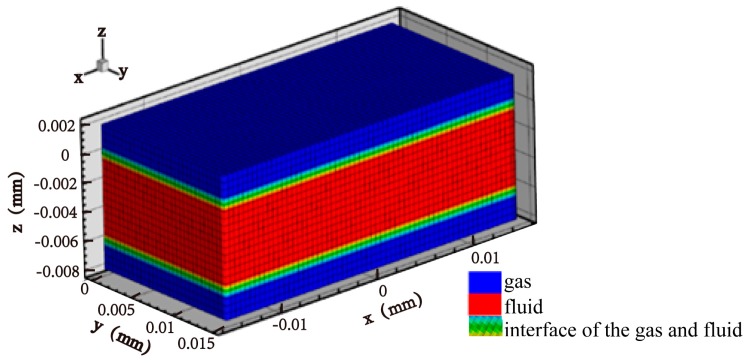
Computational domain for VPPAW model.

**Figure 6 materials-13-00303-f006:**
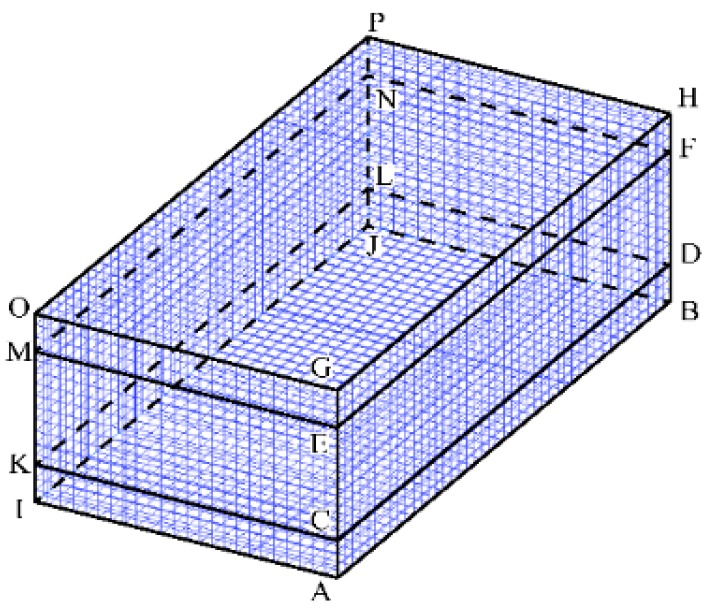
Schematic diagram of the boundary conditions.

**Figure 7 materials-13-00303-f007:**
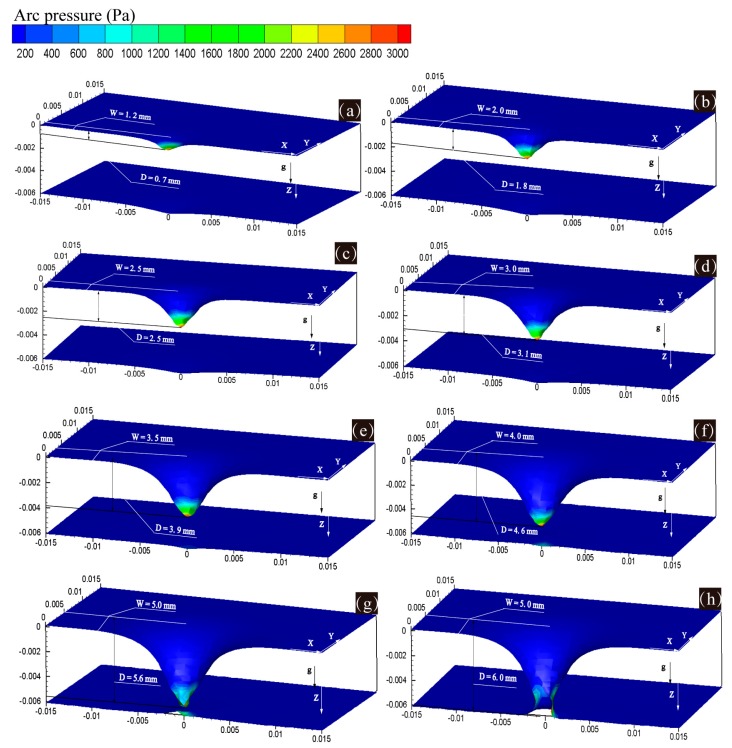
Dynamic evolution of keyhole size and arc pressure distribution on the keyhole surface of case a (**a**) 1.72 s, (**b**) 2.43 s, (**c**) 2.87 s, (**d**) 3.32 s, (**e**) 3.65 s, (**f**) 3.92 s, (**g**) 4.34 s, (**h**) 4.55 s.

**Figure 8 materials-13-00303-f008:**
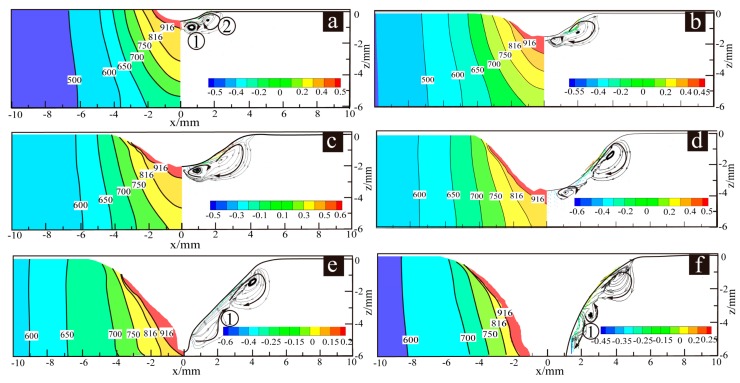
Longitudinal sectional of temperature and velocity distribution during keyhole formation process of case a (**a**) 1.72 s, (**b**) 2.43 s, (**c**) 3.32 s, (**d**) 3.65 s, (**e**) 4.34 s, (**f**) 4.55 s.

**Figure 9 materials-13-00303-f009:**
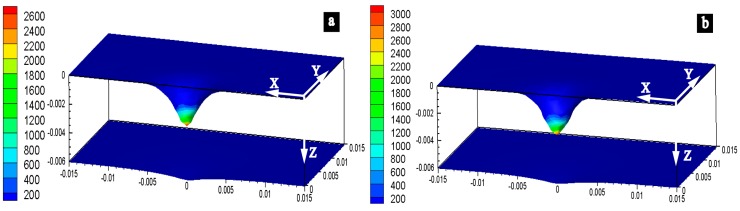
The arc pressure value in (**a**) direct current electrode negative (DCEN) phase and (**b**) direct current electrode positive (DCEP) phase obtained by simulation.

**Figure 10 materials-13-00303-f010:**
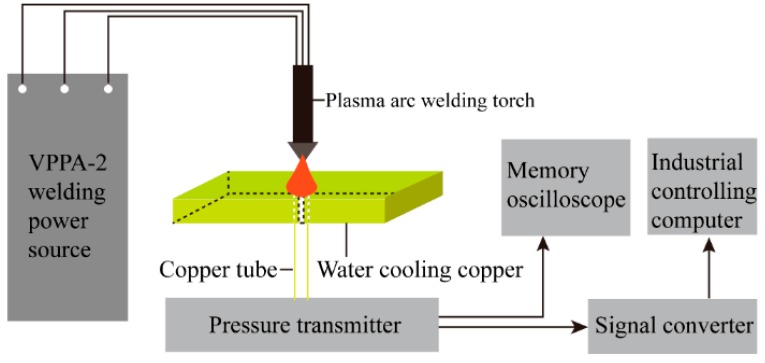
Sketch of plasma arc pressure measure and data acquisition system.

**Figure 11 materials-13-00303-f011:**
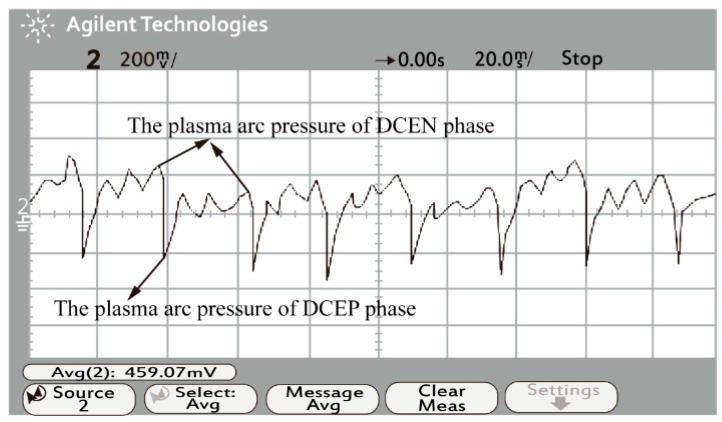
Sketch of the arc pressure waveform.

**Figure 12 materials-13-00303-f012:**
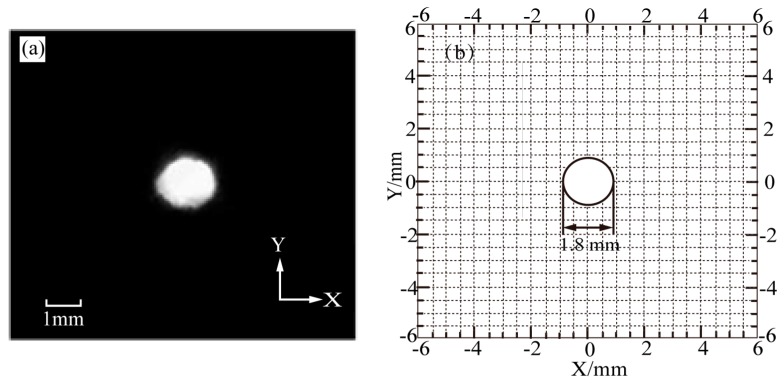
Comparison between experiment image and calculated result of the keyhole geometry of case a. (**a**) Experimental image. (**b**) Calculated result.

**Figure 13 materials-13-00303-f013:**
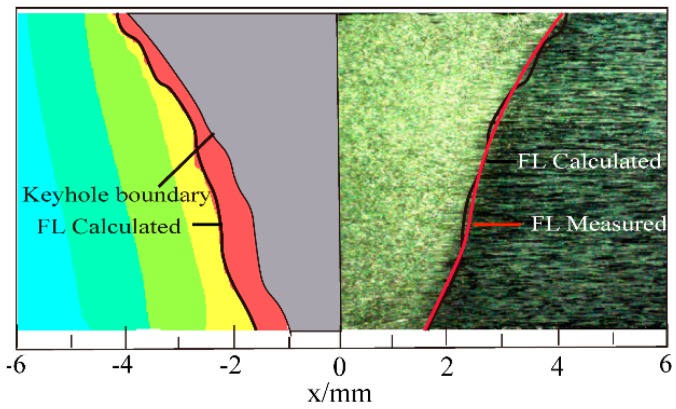
Comparison of the calculated and measured fusion line of case a.

**Figure 14 materials-13-00303-f014:**
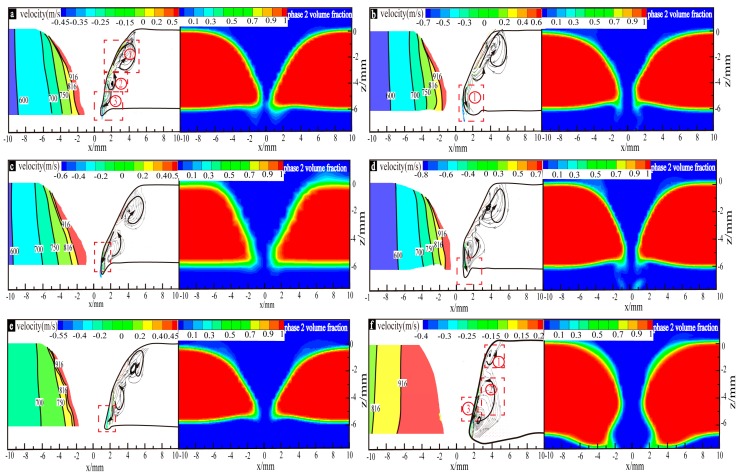
Weld pool flow and phase distribution in the critical state of the keyhole perforation for case (**a**–**f**).

**Figure 15 materials-13-00303-f015:**
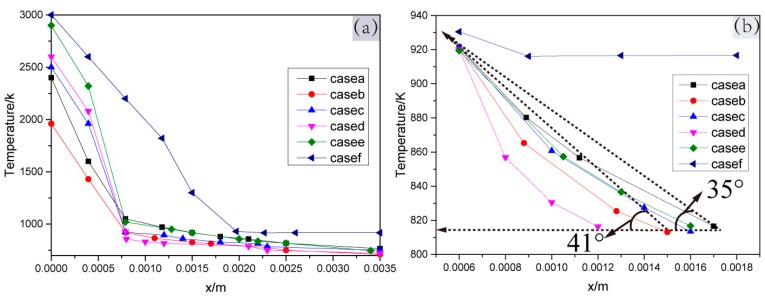
The temperature of the weld pool in the critical state of the keyhole perforation for case a-f. (**a**) Temperature gradient from 0 mm to the weld center. (**b**) Temperature gradient from 6 mm to the weld center.

**Figure 16 materials-13-00303-f016:**
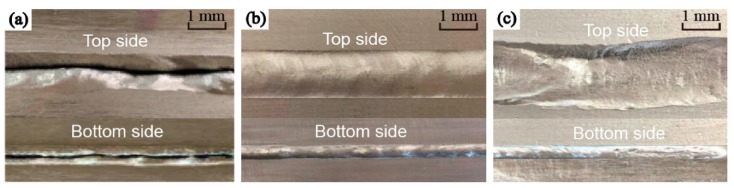
Schematic diagram of weld formations. (**a**) The weld was cut. (**b**) The weld formed well. (**c**) The weld collapsed.

**Table 1 materials-13-00303-t001:** Chemical compositions of AA2219 [25] (percent by weight).

Si	Fe	Cu	Mn	Mg	Zn	Ti	V	Zr	Al
0.2	0.3	6.58	0.3	0.02	0.1	0.1	0.05	0.2	Remainder

**Table 2 materials-13-00303-t002:** (**a**) Variable polarity plasma arc welding (VPPAW) process parameters; (**b**) VPPAW process parameters.

(**a**)			
**Parameter**		**Value**	**Remark**
$I_{0\;}$(A)		80	preheating current
$t_{0}\;$(s)		1	preheating time
$t_{s}$(s)		0.8	current rise time
$I_{\mathrm{DCEN}\;}\left(A\right)/I_{\mathrm{DCEP}\;}\left(A\right)$		110/150	case a and case b
$I_{\mathrm{DCEN}\;}\left(A\right)/I_{\mathrm{DCEP}\;}\left(A\right)$		120/160	case c and case d
$I_{\mathrm{DCEN}\;}\left(A\right)/I_{\mathrm{DCEP}\;}\left(A\right)$		130/170	case e and case f
$t_{\mathrm{DCEN}}\;$(ms)		21	
$t_{\mathrm{DCEP}}$ (ms)		4	
Flow rate of shielding gas (L/min)		15	
Arc voltage (V)		21	
Nozzle diameter (mm)		3.2	
Distance from torch to workpiece (mm)		6	
(**b**)			
**case**	$I_{\mathrm{DCEN}\;}\left(A\right)$	$I_{\mathrm{DCEP}\;}\left(A\right)$	**Q (L/min)**
a	110	150	2.4
b	110	150	2.8
c	120	160	2.8
d	120	160	3.2
e	130	170	2.4
f	130	170	2.0

**Table 3 materials-13-00303-t003:** Conditions.

Boundary	Momentum	Thermal Energy
EFNM	$\vec{v}=0$	$k*\left(\partial T/\partial \vec{n}\right)=-h_{c}\left(T-T_{0}\right)-\sigma \varepsilon \left(T^{4}-T_{0}^{4}\right)$
CDLK	$\vec{v}=0$	$k*\left(\partial T/\partial \vec{n}\right)=-h_{c}\left(T-T_{0}\right)-\sigma \varepsilon \left(T^{4}-T_{0}^{4}\right)$
CEKM	$\vec{v}=0$	$k*\left(\partial T/\partial \vec{n}\right)=-h_{c}\left(T-T_{0}\right)$
KLNM	$\vec{v}=0$	$k*\left(\partial T/\partial \vec{n}\right)=-h_{c}\left(T-T_{0}\right)$
DFNL	$\vec{v}=0$	$k*\left(\partial T/\partial \vec{n}\right)=-h_{c}\left(T-T_{0}\right)$
ABHG	$\partial u/\partial \vec{y}=0,v=0,\;\partial w/\partial \vec{z}=0$	$\partial T/\partial \vec{y}=0$

where $\partial T/\partial \vec{n}$ is the temperature gradient; $\partial T/\partial \vec{y}$ is the variance ratio of temperature along the y axes; and $\partial u/\partial \vec{y}$ and $\partial w/\partial \vec{z}$ are the velocity variation rates along the *y* and *z* axes, respectively.

**Table 4 materials-13-00303-t004:** Thermo-physical properties of AA2219 [17].

Parameter	Symbol	Number
Solid temperature	$T_{s}$	816 K
Liquid temperature	$T_{l}$	916 K
Boiling temperature	$T_{b}$	2740 K
Latent heat of fusion	$L_{m}$	391,000 J/(kg·K)

**Table 5 materials-13-00303-t005:** The comparison of keyhole perforation time, the front and back dimensions of the weld pool and the back dimensions of the keyhole.

Case	Measured/Calculated				|Relative Error|/%			
	① The Keyhole Perforation Time (t)	② The Front Dimensions of the Weld Pool (mm)	③ The Back Dimensions of the Weld Pool (mm)	④ The Back Dimensions of the Keyhole (mm)	①	②	③	④
a	4.43/4.55	4.50/4.58	1.85/1.93	0.91/0.93	2.18	2.17	4.15	3.19
b	4.32/4.28	4.05/4.15	1.85/1.90	1.00/1.05	0.90	2.41	2.63	3.80
c	4.20/4.12	4.52/4.56	2.05/2.10	1.12/1.15	1.94	0.87	2.38	2.61
d	4.08/4.05	3.88/4.05	1.55/1.60	0.78/0.82	0.49	4.19	3.12	4.87
e	3.96/4.03	5.54/5.28	2.55/2.62	1.88/1.95	1.70	4.92	2.75	3.58
f	5.15/5.21	6.35/6.68	7.20/7.45	1.54/1.62	1.20	4.94	3.47	4.93

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
