# Peer review of "Prediction of the Weld Pool Stability by Material Flow Behavior of the Perforated Weld Pool"

_materials, 2020, doi:10.3390/ma13020303_

Round 1

Reviewer 1 Report

The topic of the manuscript entitled “Prediction of the weld pool stability by material flow behavior of the perforated weld pool” falls within the scope of Materials. The paper contains very interesting theoretical considerations, numerical simulation results, experimental verification and corresponding analyses. It is of sufficient scientific interest and has originality in its technical content to merit publication. The authors have cited the relevant literature. Methods and interpretations of results are correct and novel. The issues were well presented. The arrangement of work maintains substantive continuity and constitutes a logical whole. However, the manuscript is not suitable for publication. The manuscript contains a lot of errors, mainly editorial.

Comments and remarks are presented below.

The manuscript text should be corrected and should be in accordance with Guide for Authors (Template), for example:

- font size,

- justification of equations (each equation in a separate line), see equations 5, 6, 11 and equation in 197 line of the manuscript (page 7, second line from the bottom),

- units should comply with the SI system – eg. equation (18), the next equation (line 195) has no number.

Equation (12) is not clear, I suggest e.g.: 0 for T<Ts, 1 for T>Tl, etc. In analogy to equations (17), (18) and (19).

Authors should carefully check the manuscript text.

In my opinion, the Nomenclature should be before the Introduction.

Abbreviations used for the first time in the text should be clearly explained, e.g. DCEN, DCEP.

The chemical composition of AA2219 aluminum alloy should be presented in the table and the source of this information should be indicated.

Author Response

Point 1: The manuscript text should be corrected and should be in accordance with Guide for Authors (Template), for example:

- font size,

- justification of equations (each equation in a separate line), see equations 5, 6, 11 and equation in 197 line of the manuscript (page 7, second line from the bottom),

- units should comply with the SI system – eg. equation (18), the next equation (line 195) has no number.

Response 1: I have revised the manuscript according to your comment.

Point 2: Equation (12) is not clear, I suggest e.g.: 0 for T<Ts, 1 for T>Tl, etc. In analogy to equations (17), (18) and (19).

Response 2: I have revised equation (12) as shown in line 133 in the revised manuscript.

Point 3: The Nomenclature should be before the Introduction.

Response 3: I have changed the Nomenclature before the Introduction.

Point 4: Abbreviations used for the first time in the text should be clearly explained, e.g. DCEN, DCEP.

Response 4: DCEN and DCEP are the direct current electrode negative and direct current electrode Positive phase respectively in variable polarity plasma arc welding (VPPAW). The abbreviation PAW is plasma arc welding.

Point 5: The chemical composition of AA2219 aluminum alloy should be presented in the table and the source of this information should be indicated.

Response 5: The base metal employed in this study was AA2219 aluminum alloy plates with a chemical composition as shown in Table 1.

Table 1. Chemical compositions of AA2219 aluminum alloy [25] (percent by weight).

Si

Fe

Cu

Mn

Mg

Zn

Ti

V

Zr

Al

0.2

0.3

6.58

0.3

0.02

0.1

0.1

0.05

0.2

Remainder

Reviewer 2 Report

In this work the authors presents the application of a computational fluid dynamics model for thermo-mechanical coupling simulation of the weld pool used in variable polar plasma arc welding. The research appears to be efficiently done and appropriately reported, however the standard of English must be improved. Nevertheless, there some questions and corrections that must be answered to improve and complete the document.

Abstract: At least a half of the abstract is a copy past of conclusions, even the mistakes. Please, rewritten the abstract avoiding the copy past of other sections of the manuscript.

The authors must correct the manuscript formation they changed the font size throughout the text, as for example in the lines: 1, 2, 63, 64, 67-69, 75-79, 99, 110, 111, 149, 150, 179, 180, 202, 203, 238, 244, 249, 251, 269, 270, 311, 326, 363, 364, 370-373, 388, 389, 395, 400, 401, 414, 415, 420-422, 433-437, 441-443, 448, 449.

Line 31: Please, change the sentence “… at low temperature and high temperature, …” to “… at low and high temperatures, …”

Line 33: Please change the sentence “… great application potential in …” to “… great potential application in …”.

Line 33: The authors claim that VPPAW has a great potential application in “the welding of medium-thickness aluminum alloy”, however I suggest an indication of a range of thicknesses values, ex. (xx mm, yy mm).

Line 46: The first time that the authors use an abbreviation they must indicate the meaning of that. So, the abbreviation PAW must be explained.

Lines 85-86: I suggest a table to indicate the chemical composition.

Lines 126-127: Please, the representation of equations (5) to (6) must be improved.

Please, change the word “formula” to “equation” throughout the text, is scientifically more correct word.

Lines 146-150: Please, the representation of equations (9) to (11) must be improved.

Lines 361, 362 and 382: Please, change “pa” to “Pa”.

Line 372: Please change the word “vppa-2” to “VPPA-2”.

Section 5 (results and discussion): the experimental procedure is not very clear and is described very superficially. Please, describe the experimental procedure in more detailed way. It is impossible to compare the numerical and experimental results with the presented information in this manuscript.

Line 456: Please change “velocities 0.45 m/s ,0.55 m/s ,0.60 m/s, the …” to “velocities 0.45 m/s, 0.55 m/s, 0.60 m/s, the …” and “… stably. when the …” to “… stably. When the …”

Lines 458-460: The sentence which begins in “The weld pools …” and finish in “… increased continuously.” Must be improved to make it clearer and more understandable.

Author Response

I'm sorry that my standard of English needs to be improved. In the past two days, I consulted with my science and technology thesis writing teacher and revised my manuscript.

Point 1: Abstract: At least a half of the abstract is a copy past of conclusions, even the mistakes. Please, rewritten the abstract avoiding the copy past of other sections of the manuscript.

Response 1: The abstract has been rewritten as shown in line 8-20 in the revised manuscript.

Point 2: The authors must correct the manuscript formation they changed the font size throughout the text, as for example in the lines: 1, 2, 63, 64, 67-69, 75-79, 99, 110, 111, 149, 150, 179, 180, 202, 203, 238, 244, 249, 251, 269, 270, 311, 326, 363, 364, 370-373, 388, 389, 395, 400, 401, 414, 415, 420-422, 433-437, 441-443, 448, 449.

Response 2: I have checked the manuscript carefully, and revised the font size.

Point 3: Please, change the sentence “… at low temperature and high temperature, …” to “… at low and high temperatures, …”. Please change the sentence “… great application potential in …” to “… great potential application in …”.

Response 3: I have revised it according to your comment as showed in line 27-28 in the revised manuscript.

Point 4: The authors claim that VPPAW has a great potential application in “the welding of medium-thickness aluminum alloy”, however I suggest an indication of a range of thicknesses values, ex. (xx mm, yy mm).

Response 4: I have indicated a range of thicknesses values of medium-thickness aluminum alloy about 4mm-20mm, and the source of this information was indicated in reference [2] as showed in line 29 in the revised manuscript.

Point 5: The first time that the authors use an abbreviation they must indicate the meaning of that. So, the abbreviation PAW must be explained.

Response 5: The abbreviation PAW is plasma arc welding as showed in line 40 in the revised manuscript.

Point 6: I suggest a table to indicate the chemical composition.

Response 6: The workpiece employed in this study was AA2219 aluminum alloy plates with a chemical composition as shown in Table 1 in the revised manuscript. The source of this information was indicated in reference [25] as showed in line 75 in the revised manuscript.

Table 1. Chemical compositions of AA2219 aluminum alloy [25] (percent by weight).

Si

Fe

Cu

Mn

Mg

Zn

Ti

V

Zr

Al

0.2

0.3

6.58

0.3

0.02

0.1

0.1

0.05

0.2

Remainder

Point 7: Please, the representation of equations (5) to (6) must be improved. Please, change the word “formula” to “equation” throughout the text, is scientifically more correct word. Line 146-150: Please, the representation of equations (9) to (11) must be improved. Lines 361, 362 and 382: Please, change “pa” to “Pa”. Line 372: Please change the word “vppa-2” to “VPPA-2”.

Response 7: I have improved equations (5) to (6) and (9) to (10) as showed in line111-112 and line127-128 respectively, and changed the word “formula” to “equation” throughout the text, and revised “pa” to “Pa” and “vppa-2” to “VPPA-2” as showed in line332 and line342 respectively.

Point 8: Section 5 (results and discussion): the experimental procedure is not very clear and is described very superficially. Please, describe the experimental procedure in more detailed way. It is impossible to compare the numerical and experimental results with the presented information in this manuscript.

Response 8: I'm so sorry that I haven't describe section 5 clearly. Please allow me to explain the development of this part again.

Firstly, with the help of VPPA welding system as shown in Figure 1 and fluent software, the welding experiments and simulation tests were carried out respectively with the welding parameters of case a as shown in Table 2. The arc pressure distribution on the surface of the weld pool, the keyhole size and the shape of the fusion line were obtained from simulation and experiment respectively. Because I think that it is necessary to verify the model to ensure the accuracy of the subsequent analysis results, therefore, the accuracy of the model was verified by comparing the simulation and experimental results.

Then, for case b-f, simulation tests and welding experiments were carried out with the help of VPPA welding system and fluent software used the welding parameters of case b-f as shown in Table 2.The flow behaviors of the weld pools and its causes were analyzed in detail, and the weld pools stability were predicted by its three kinds of flow behaviors. Since the stable existence of the weld pool is the necessary for a good weld formation [1] (The order of this reference is independent of the order of references in the manuscript), the predicted stability of the weld pools was further proved by different weld formation results. Furthermore, the reliability of predicting the stability of the weld pool by the material flow behaviors of the weld pools was verified.

[1]Y, C, Lei. Study on the weld formation in vertical keyhole plasma arc welding[J]. Journal of Jiangsu University (Natural Science Edition), 1994(04), 62-67.

Point 9: Line 456: Please change “velocities 0.45 m/s ,0.55 m/s ,0.60 m/s, the …” to “velocities 0.45 m/s, 0.55 m/s, 0.60 m/s, the …” and “… stably. when the …” to “… stably. When the …”

Response 9: I have revised the paragraph as follows as shown in Line 405-409 in the revised manuscript.

The weld pool can exist stably when the material in the bottom of the weld pool can form confluences with the maximum flow velocities 0.45 m/s ,0.55 m/s and 0.60 m/s. The weld pools were cut when the material flow downward and outward with the maximum velocities 0.7 m/s and 0.8m/s, metal gaps appeared according to the law of mass conservation.

Point 10: Lines 458-460: The sentence which begins in “The weld pools …” and finish in “… increased continuously.” Must be improved to make it clearer and more understandable.

Response 10: I have revised the paragraph as follows as shown in Line 409-411 in the revised manuscript.

When the maximum material flow velocity was 0.40 m/s, there was metal refluxing phenomenon in the weld pool for a long time, which made the molten metal increased. The gravity of weld pool increases with the amount of molten metal, so the weld pool collapsed downward under the action of larger gravity.

Round 2

Reviewer 2 Report

The second version of manuscript improved significantly when compared with first version. So, in my opinion the manuscript can be accepted for publication. However, in table 2 occurred any mistake when the authors saved the document in pdf version (appears a square symbol in place of letters), please be attention to this table before the publication.
